# Exploring the relationship between regional tourism development and land use efficiency: A case study of Guangxi Zhuang Autonomous Region, China

Qiuli Meng, Hongwen Pi *, Tao Xu, Lihua Li

School of Management, Guangxi Minzu University, Nanning, China

* pihongwen@stu.gxmzu.edu.cn

## Abstract

The utilization efficiency of land resources is an essential embodiment of economic development, social development, and ecological development and is a critical core to measure how to maximize the efficiency of land resources under limited conditions. The land is an important content and essential carrier of the research of tourism development level. This paper selects Panel Data from 2010 to 2019 to research the Guangxi regional tourism development. The entropy weight method and stochastic frontier production function (SFA) model were used to evaluate the development level of urban-rural tourism and the utilization efficiency of land resources in Guangxi. This paper uses the Panel Vector Autoregression (PVAR) model to analyze the internal relationship between urban-rural tourism development. The results show that: (1) Guangxi has a good level of tourism development and a high land use efficiency. (2) There is a reciprocal causation relationship between the regional tourism development level and land use efficiency in Guangxi, with significant levels of 0.005 and 0.034 respectively, indicating high credibility. This indicates that there is a mutual promotion and interaction between the two, which rely on and drive each other, promoting the joint sustainability of tourism development and land use efficiency. (3) . The tourism development level is greatly influenced by itself, with impact values all above 0.99. At the same time, land use also has a significant self-impact, with impact values all above 0.87. Their internal optimization system is solid and endogenous impetus is robust, which can drive their development. Establishing an effective strategy for developing and protecting land use is beneficial to promote the long-term effectiveness of sustainable tourism development, enhancing high-quality development of the tourism economy and improving people's living standards and quality.

**Data Availability Statement:** All data files areavailable from the Figshare database (doi: https://doi.org/10.6084/m9.figshare.24126672)

## 1. Introduction

As a new pillar industry, tourism has become an essential direction for development and investment in many cities. In urbanization, tourism development has become one of the

**Funding:** This article was financially supported by' the National Office for Philosophy and Social Sciences: Research on Tourism Cultural Space Production and Social Governance Innovation in southwest minority traditional villages (20BJY210)'. The funders had no role in study design, data collection and analysis, decision to publish, or preparation of the manuscript.

critical driving forces. However, there is still a waste of tourism land resources [1], with more vacant and wasteland, such as lousy tourism development, overgrown scenic spots, and little planning and use. In the third national land survey, the service, change, utilization efficiency, and protection degree of land resources were detected and mapped by 3S Technology. The survey results show that some areas still have large amounts of in-efficient and unused land [2]. Land use planning should be vigorously adjusted to give full play to the importance of land in tourism and promote high-quality and efficient tourism development. Tourism development and construction require a large amount of land for building tourist attractions, hotels, and other related facilities. If the land use efficiency in the research area for tourism development is not high, it is very likely to damage the land quality and other useful values of the land. Therefore, studying the relationship between the tourism development level and land use efficiency is of great value.

In 2019, the Central Committee of the Communist Party of China (CPC) explicitly proposed the "three lifestyles" (production, living, and ecology) spatial master plan in its Opinions on Establishing a Territorial Spatial Planning System and Supervising its Implementation to strictly control land for urban and rural planning and improve the efficiency of land use. According to the national policy of 'Three Zones and Three Lines', Guangxi has formulated its own 'Three Zones and Three Lines' policy, which aims to improve the national land spatial planning, enhance the quality and value of land utilization, and ensure the reasonable development of cultural and tourism industries. In November of the same year, Guangxi Natural Resources Bureau issued a policy on land use to support the high-quality development of cultural tourism. We should reduce the cost of land use, guarantee the space of land use, make full use of rural collective land, and support the development of high-quality cultural tourism. Government policies will continuously address various land use issues that arise in the development of tourism. Not only that, but tourism development is a process of ecological protection and rapid economic operation, and the development of regional tourism is conducive to the promotion of the local economy and the improvement of people's living standards. Limited land resources have become an important element in the development of tourism. Tourism areas combine local human development and tourism activities, which not only enhance the value of land use [3] but also enrich tourism connotation [4], drive the extension of the tourism industry, and protect the ecological environment [5]. As an important component of tourism resources and a carrier for tourism development, land resources cannot be ignored. Therefore, in the study of the entire tourism development process in Guangxi, our research team further investigated the utilization of tourism land resources.

## 2. Literature review

The relationship between land use and tourism was proposed in the early 1930s. The publication of Mc Murry's article 'The relationship between tourism and land use' [6] has received more attention from national and international scholars and has led to a greater awareness of the benefits of tourism development. This also highlights the importance of land use [7]. The essence of land use is to enhance and maintain the current function of land use [8], and the extent of land use is directly linked to the region's sustainable development and the building of an ecological civilization [9]. The idea of sustainable land use was explicitly put forward in Agenda 21 in 1992 [10], which provided a central point for policy formulation and a new direction for academic research, making land issues a popular study area [11]. Tourism, as a typical form of land use activity, is one of the industries that rely on land resources. Land resources are the basic conditions and important resource support for the development of tourism.

Improving land use efficiency is a crucial way to promote sustainable land use and has become an essential research topic for scholars and experts. The current methods for calculating land use efficiency are the Data Envelopment Analysis method (DEA) and the stochastic frontier production function method (SFA) [12, 13]. DEA is mainly based on the linear programming method to measure, and SFA is based on the production function to calculate the unit efficiency [14]. Compared with the calculation and application of the efficiency of SFA in the tourism industry, the DEA model does not separate the compound interference term (random interference term and inefficiency term), which results in the instability of the calculation results [15]. However, many scholars still use this research method to study tourism and land use efficiency and contribute significant results to scientific research. In the study of urban tourism efficiency, dabble in Yangtze River Delta [16], Beijing-Tianjin-Hebei Urban agglomeration [17], Europe [18], Guangxi Autonomous Region [19], and so on; In the specific content of tourism research, including Forest Park [20], service quality [21], poverty alleviation tourism [22], tourism industry [23] and so on. In land use efficiency, Zhu XuSen studied its economic and ecological efficiency through the DEA model [24], and Zhu Qiaoxian studied carbon emission measurement [25]. However, compared with DEA, SAF can keep the stability of the results of data calculation by stripping out the compound interference term. The measurement error, the uncertainty of the economic environment [26], and technical inefficiency [27] in developing countries are more pronounced. Therefore, it is more suitable to use the SFA model to measure the development efficiency of developing countries. Li Liang and Zhao Lei use the stochastic frontier to analyze China's tourism development efficiency. The result shows that China's tourism development efficiency has been improving year by year [28]. Dilawar Khan [29] used SFA to study the relationship between energy efficiency and ecological footprint and believed that efficient utilization of energy resources as well as investment in agriculture are necessary for a sustainable environment. Meanwhile, Professor Dilawar Khan also studied the impact of technological innovation on energy efficiency [30]. Zainab Bibi studied the technical and environmental efficiency of the agriculture sector in South Asia [31] and suggested that the agriculture sector should prioritize collaboration in research and development.

Many scholars use the DEA model to calculate land use efficiency, industrial development efficiency, etc. Few scholars use SFA to measure land use efficiency, which is very rare in the field of tourism, and it can make up for the limitation of the DEA model and separate the compound disturbance term. Therefore, this paper mainly uses the SFA model to calculate the land-use efficiency of Guangxi and analyzes the internal relevance with the level of tourism development.

## 3. Data

### 3.1 Study area

The geographical range of the Guangxi Zhuang Autonomous Region is 20˚ 54'-26˚ 24' N and 104˚ 26'-112˚ 04' E in the southeast of our country (Fig 1). It includes 14 municipalities with Nanning as their capital. According to the Guangxi statistical yearbook statistics, Guang-xi's land area changed little between 2010 and 2019, with 237,771 square kilometers in 2019. With the increase in cultural tourism land area, the classification management of cultural and tourism land, the adoption of flexible land supply mode, and the reduction of land costs for cultural and tourism purposes have been achieved. Therefore, it is encouraged to revitalize and utilize collective rural land, wasteland, etc., increase the area of cultural and tourism land, and improve land reuse efficiency. With the rapid development of the regional economy, total economic income increased from 935.651 billion yuan in 2010 to 2127.357 billion yuan in 2019,

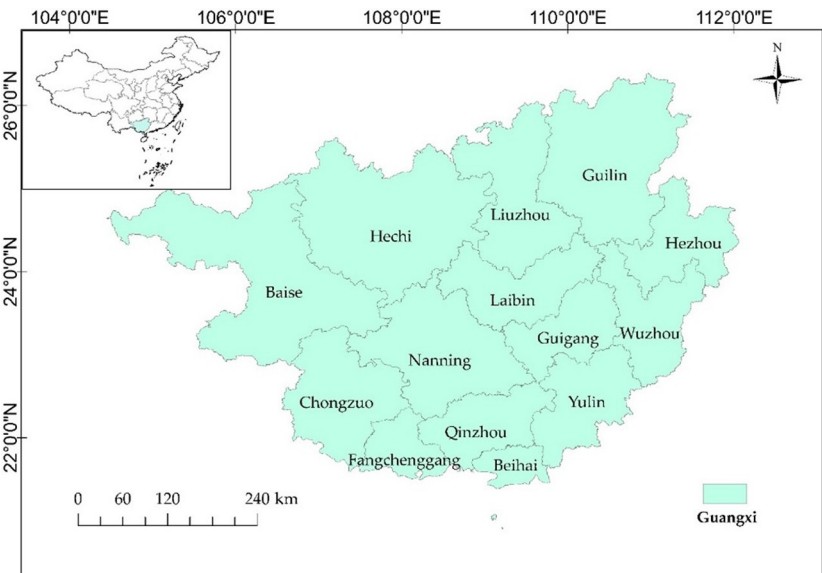

**Fig 1. Guangxi administrative division.** Note: Based on the standard map production of the National Natural Resources Department's Standard Map Service website (GS(2019)1822), the base map boundaries have not been modified. (http://bzdt.ch.mnr.gov.cn/download.html?searchText=GS(2019)1822).

of which tourism income increased from 95.377 billion yuan to 1024.144 billion yuan, the share of total revenue increased from 10.19% to 48.16%. The data shows that the rapid development of tourism makes tourism a vital pillar industry for regional economic development. The number of scenic spots has increased exponentially, and with the development of the tourism industry, the number of tourism organizations is rising. Rich tourism resources promote the integrated development of the tourism industry,to enhance the overall economic development of Guangxi region.

## 3.2 Data sources

This paper studies the intrinsic links between regional tourism development and land use in Guangxi. The data needed should objectively reflect the development trend of the statistical year. Therefore, data collection considers the objective reality of the data, comparability, and finally, the PVAR model of the data length requirements. The final data were selected from Guangxi statistical yearbook, the Economic and Social Development Statistical Bulletin, and China Statistical Yearbook from 2011 to 2020. However, the data on employees in the first industry and total fixed asset investment in 2018 and 2019 are missing, so the missing data have been completed utilizing interpolation. From 2003 to 2010, statistics on the treatment rate of industrial wastewater and the rate of environmental protection treatment of industrial waste gas show that the compliance rate was above 0.9, and the data from 2011 to 2019 are missing. With the progress of science and technology, waste gas and wastewater treatment has a certain strict standard. For the convenience of statistics and without affecting the result of statistics, the rate of reaching the standard is set to 1. In addition, the amount of investment in industrial pollution and the value of standard treatment are the total data of autonomous regions. To reduce the error of the total data, it is convenient to calculate the development of Guangxi. In this paper, the total data will be divided according to the proportion of data to Guangxi's large industrial enterprises as the basis.

## 4. Methodology

### 4.1 Index construction and description

**4.1.1 The index selection of Guangxi regional tourism development level(X).** In selecting indicators for tourism development level (Table 1), this paper draws on the value evaluation system and measuring indicators studied by scholars Fu Haichen [32] and Zhang Guanghai [33]. Total tourism consumption, gross domestic product (GDP), and gross tertiary sector of the economy product (GDP) were selected to measure the benefits of tourism in the development process and its contribution to the overall economy. To some extent, a regional tourism destination's development process and level can be measured by selecting the total amount of tourism reception, the number of scenic spots above A story, and the number of tourism organizations. The statistical measurement of highway network density, population density, per capita area of green park space, and built-up area indicates the basic development process of tourism.

**4.1.2 The index selection of land use efficiency(Y).** Land use efficiency can be regarded as the benefit of developing the social economy and environment. In this paper, the cobb-douglas production function algorithm is used for reference. Land, capital, and labor force are the input factors, taking the total industrial development as an output factor. The development of society requires the input of labor and capital, with land serving as the carrier of space, and the value generated has certain measurement standards in economic development. Negative output that affects social and economic development needs to be eliminated First, the input value and output benefits of land use are calculated, with the built-up area, investment in fixed assets, primary industry, secondary sector of the economy, and tertiary sector of the economy labor as inputs [34, 35], taking the gross output of the primary, secondary sector of the economy and tertiary sector of the economy industries as the expected output value. Secondly, in this article, the output of industrial wastewater treatment, industrial waste gas treatment, and industrial solid waste treatment in the governance of the "Three Industrial Wastes" [36] are taken as non-expected values. After accounting for the expected value, the stochastic frontier model calculates the land use efficiency.

### 4.2 Methods of data processing and measurement

**4.2.1 Data processing of regional tourism indicators.** In this paper, the panel data are normalized and weighted by the entropy weight method. Finally, the tourism development level of each region in different periods is calculated by calculating the weight value:

(1) Calculate the information entropy of the index

$$T_{ij} = -\ln{(n)}^{-1} \sum\nolimits_{1=1}^{n} k_{ij}\, lnk_{ij} \tag{1}$$

(2) Calculate the weight of the indicator

$$W_i = \frac{1 - T_i}{n - \sum T_i}(i = 1, 2, 3\ldots., n) \tag{2}$$

(3) Level of regional tourism development

$$\text{X} = X_1 W_1 + X_2 W_2 + \cdots..X_n W_n \tag{3}$$

In the whole calculation process, n is the number of years, $k_{ij} = \frac{X_{ij}}{\sum_{i=1}^{n} X_{ij}}$, when $k_{ij} = 0, \underset{P_{ij} \to 0}{Lim}\, k_{ij} lk_{ij} = 0$, Where $X_{ij}$ Is the value of the j Index in the I year after the normalization

**Table 1. Evaluation index system of tourism development level in Guangxi.**

| Indicator name | Units | Definition | Indicator description |
|---|---|---|---|
| Gross Domestic Product ($X_1$) | Billion | Status of regional economic development | Reflecting the size of regional economic returns, the more significant the income, the better the economic development benefits. It is more able to promote tourism development. |
| Gross tertiary sector of the economy product ($X_2$) | Billion | Regional tertiary sector of the economic development | The consumer is the primary tertiary sector of the economy of income. When the gross domestic product is more prominent, indicating that the consumer spending capacity is more substantial, the greater the willingness to spend |
| The total cost of travel ($X_3$) | Billion | The development of regional tourism | Reflecting the consumption of tourists in the region, when the more significant number means that tourists spend more, tourism income is higher. |
| The total amount of tourist reception ($X_4$) | Ten thousand person-time | Total number of domestic and foreign tourists | Reflecting the total number of tourists to the scenic spot and the degree of attraction of regional tourism, the greater the number indicates the lure of the scenic spot. |
| Number of A-class scenic spots ($X_5$) | A | Total number of A-5A scenic spots in the area | The greater the number of A-class scenic spots, the more destinations tourists can visit, promoting consumption growth. The higher the level of a-level scenic spots, the more the number, indicating that the development of tourism destination function is sound and perfect, more attractive to tourists to play. |
| Number of tourist agencies ($X_6$) | A | Number of management departments + number of star hotels + number of travel agencies | Reflecting the strength of tourism logistics support and convenience, the higher the value is conducive to tourism development. |
| The density of the highway network ($X_7$) | km / ten thousand square kilometers | Length of city km/area of land | Reflecting the degree of tourist travel traffic convenience, the greater the numerical value indicates the degree of comfort is higher. |
| Population density ($X_8$) | Person/km$^2$ | Total urban and rural population/area of land | Reflect the thickness of the urban population |
| Per capita park green space area ($X_9$) | M$^2$/person | The area of green space in an urban area/ the number of urban population | Reflecting the degree of urban greening construction, the higher the numerical value, the better the urban green development |
| Green area covered by the built-up area ($X_{10}$) | Percentage (%) | Built-up area green area/built-up area | |

treatment., $W_i, X_i$ respectively represent the calculation weights of indicators and the level of tourism development in year i.

**4.2.2 Data processing of land use efficiency index.** In this paper, the stochastic frontier production function is used to calculate the land use efficiency, so the following stochastic frontier model is constructed:

$$\ln(Z_{it} - Q_{it}) = \alpha_0 + \alpha_1 \ln D_{it} + \alpha_2 \ln B_{it} + V_{it} - U_{it} \qquad (4)$$

In the above formula, $Z_{it}$ It is the sum of the value of the built-up land for the primary industry, secondary sector of the economy, and tertiary sector of the economy units. D and B are the total amount of investment in fixed assets per unit of built-up area and the total amount of labor in the three industries per unit of built-up area. $V_{it}, U_{it}$ Denote random error term and production inefficiency time, respectively. $Q_{it}$ It is the non-expected output value of the built-up area. The construction area per unit calculates the total industry investment and the treatment efficiency of the three wastes. The method of calculation is as follows:

$$Q_{it} = \left(\frac{E_{it}}{e_{it}} + \frac{c_{it}}{c_{it}} + \frac{H_{it}}{h_{it}}\right)/S_{it} \qquad (5)$$

In the above, $E_{it}, C_{it}, H_{it}$ Denotes the investment quota of industrial wastewater treatment, industrial waste gas treatment, and industrial waste solid treatment in region i in the t-year. $e_{it}, c_{it}, h_{it}$ Indicate the rate of industrial wastewater discharge, the comprehensive utilization of

industrial waste solid, and the rate of environmental treatment of industrial waste gas in the tourist area i in the t year. *four* Denotes the built-up area of i in year t.

Therefore, the expression of comprehensive land use efficiency is as follows:

$$Y_{it} = e^{-U_{it}} \tag{6}$$

## 4.3 Build the PVAR model

The PVAR model can analyze the interaction between endogenous variables in panel data, which not only effectively solves the problem of individual heterogeneity, but also fully considers the individual and time effects. This model can solve the econometric problems involved in different fields. Therefore, in this paper, the PVAR model is used to analyze the relationship between the level of regional tourism development and land use efficiency in Guangxi. This model is less restrictive to the time series, can effectively analyze the variables in time, and has a high KMO value. The PVAR model in this paper mainly refers to the construction method of Zhang Helin, Wang Yachen, and Liu Ying [37]. It uses Granger causality. It maps its change process through GIS. The impulse response graph is used to analyze the impulse response among the variables to analyze the internal fluctuation degree of land use efficiency and tourism development level. In the following formula, Z represents a two-dimensional column matrix composed of Guangxi tourism development level (X) and land use efficiency (Y). i represents the selected region, t represents the time series, Zit and Zit-h represent the two-dimensional column matrix between them, and the two-dimensional column matrix lags the h-th order. $\partial_i$, $\varepsilon_i$, $u_0$, $\varphi$, $u_j$ denotes individual fixed-effect column, time-effect column, intercept term vector, the number of lag stages, the parameter matrix of the j-order lag, and $\beta_{it}$ It is a random interference term with normal distribution.

$$Z_{it} = \partial_i + \varepsilon_i + u_0 + \sum_{j=1}^{\varphi} u_j Z_{it-J} + \beta_{it} \tag{7}$$

## 5. Results and discussion

### 5.1 Index construction and description

The panel's root smoothness was tested before the model analysis to avoid the data model's false regression. In this paper, the ADF-Fidher test was used. Although only the Inverse Norma value was more significant than 0.05 after the stationarity test, it had little effect on the stationarity of the overall data model. In order to ensure the robust stability of the later data analysis, the whole data are processed by second-order difference. The results of the second-order stability test are shown in the table below (Table 2). At the significant level of 1%, the assumption of unit root is rejected, so we can judge that the stability of each sequence is high, and the PVAR model can be constructed.

Before analyzing the model, we need to determine the optimal lag order of the model, which is usually determined by the principle of minimum selection value of MBIC, MAIC, and MQIC. The test results are as follows (Table 3).

According to the standards of MBIC, MAIC, and MQIC, the optimal lag order is 1. Generally speaking, MBIC, MAIC, and MQIC minima are at the same level, so they are more inclined to the simplified model. Therefore, the optimal lag order of the PVAR model is 1 order. The PVAR model of Guangxi regional tourism development level and land use

**Table 2. Panel root inspection results.**

| | | Statistic | p-value |
|---|---|---|---|
| tour | Inverse chi-squared (28) | 164.5259 | 0.0000 |
| | Inverse normal | -5.1706 | 0.0000 |
| | Inverse logit t (74) | -10.3788 | 0.0000 |
| | Modified inv. chi-squared | 18.2439 | 0.0000 |
| land | Inverse chi-squared (28) | 937.2761 | 0.0000 |
| | Inverse normal | -27.8344 | 0.0000 |
| | Inverse logit t (74) | -69.1326 | 0.0000 |
| | Modified inv. chi-squared | 121.5071 | 0.0000 |

efficiency is obtained using the GMM estimation method with the optimal lag of 1 order. The results are as follows (Table 4).

According to the results of the above table, the elastic coefficients of the lag of tourism development level and land use efficiency for Guangxi are -0.0496, 0.2889, 0.0127, and 0.1795, respectively. The elastic coefficient values were significant at 1% and 5% of the corresponding statistical discounts. To some extent, the tourism development level is related to the land use efficiency in time series, and its internal development mechanism is improved. Regional resident population, capital input, regional area, and industrial development change little quickly. To some extent, land use area and land use efficiency do not significantly impact the above factors. As a comprehensive industry, tourism is influenced by many factors, leading to the rapid change of many factors and indirectly affecting land use area and efficiency. In the lag period, the self-effect of tourism is negative but not significant, and the development level of regional tourism and land use efficiency are both significantly positive. This indicates that the level of tourism development and land use efficiency of Guangxi promote each other.

To guarantee the stability of the PVAR model unit circle test is used to verify the strength of the PVAR model by observing whether all the characteristic roots are in the unit circle (less than 1).

The above unit circle test (Fig 2) shows that the two estimated points are 0.1945 and 0.0647, respectively, all in the unit circle. The PVAR model has some stability and a long-term stable relationship between the variables. So we can continue with the Granger causality test.

## 5.2 Granger causality test analysis

The PVAR model's data shows that Guangxi's tourism development level and land use efficiency mutually influence it. Therefore, a Granger causality test was performed to verify the causal relationship between the variables X and Y. The results were as follows (Table 5). Fig 3 intuitively displays Guangxi regional tourism development level, land use efficiency change degree, degree, and trend, and cross-reference analysis with the Percy Grainger causality test.

The above test results showed that the original hypothesis was rejected. The P value passed the 5% statistical test, which showed that the level of tourism development and land use efficiency were mutually Granger causality. With close ties, the two promoted each other to achieve sustainable growth [38].

Fig 3 shows that Guangxi's tourism development level and land use efficiency gradually improved with the time series. The level of development and utilization efficiency demonstrated by the example in the figure is the level in the time zone. At the same time, due to the subjectivity of the index selection and the variability of economic and social development, its value can not fully represent the actual level of development and efficiency. However, they can

**Table 3. Effects of the optimal order of hysteresis.**

| lag | CD | J | J p-value | MBIC | MAIC | MQIC |
|---|---|---|---|---|---|---|
| 1 | 0.9999999 | 27.55111 | 0. 0064317 | -17.30093 | 3.551109 | -4.091987 |
| 2 | 1 | 22.69081 | 0. 0037847 | -7.210545 | 6.690812 | 1.595415 |
| 3 | 0.9999998 | 15.49382 | 0. 0037793 | 0.5431387 | 7.493817 | 4.946119 |

be used as a reference to show its development trends and study its inherent relevance. In combination with Fig 4, Hechi and Baise are mainly mountainous areas with higher elevations and relatively less usable land area. The utilization efficiency of these areas is earlier than that of other areas, and their development and utilization efficiency change slowly. The North Sea, Qinzhou, and Fangchenggang are mainly coastal regions with rich tourism resources and attractive tourist destinations. The development of tourism is relatively rapid, and the development of marine tourism is good. The change in land use is evident along with the time series evolution. Nanning, Laibing, and Gui-gang are in plain and basin regions with lower elevations. The land available for tourism development is broader than in mountainous areas, and the potential for tourism development is profound.

As a restrictive factor of tourism development, land is also an essential carrier of tourism development. Limited land use and tourism development space are also subject to certain restrictions. But how to improve the use efficiency in the limited use space has become an essential topic of regional tourism development and an important direction to promote the sustainable development of land and improve the efficiency of land use. Tourism development has a counter-effect on the land [7] and becomes one of the factors to boost the efficiency of land use. The story of tourism needs to expand the scope of tourism and the high-quality development of tourism to improve the efficiency of land use and proper planning of tourism development [39].

## 5.3 Impulse response analysis and variance decomposition analysis

The impulse response is based on the assumption that other factors remain constant and that the impact of the error term of one variable plus a standard deviation on the dynamic effect of another variable in the current and future periods [40]. The result of each variable on itself and other variables in the 0–10 period changes as follows, the horizontal axis represents the number of the response period, and the vertical axis represents the impact of 95% confidence intervals.

According to the pulse corresponding in the Fig 5 above, land use efficiency positively impacts the tourism development level. The positive effects of land on tourism and tourism on the ground are significant from zero to the tenth period. The first period shows a slight upward trend, and both approach zero after the second period. The impact of land use efficiency and tourism development level is not significant in the zero phases but dramatically substantial after the first phase. The fluctuation value tends to zero as time goes on. The impact of tourism

**Table 4. PVAR model results.**

| | tour | | land | |
|---|---|---|---|---|
| L1. tour | -0.0496 | (0.0444) | 0.2889*** | (0.1032) |
| L1. land | 0.0127*** | (0.0060) | 0.1795*** | (0.0364) |

Note: the standard error in parentheses*,*** is significant at 1. 0%,5%, and 1% significance levels, respectively, and L1 is a lag phase.

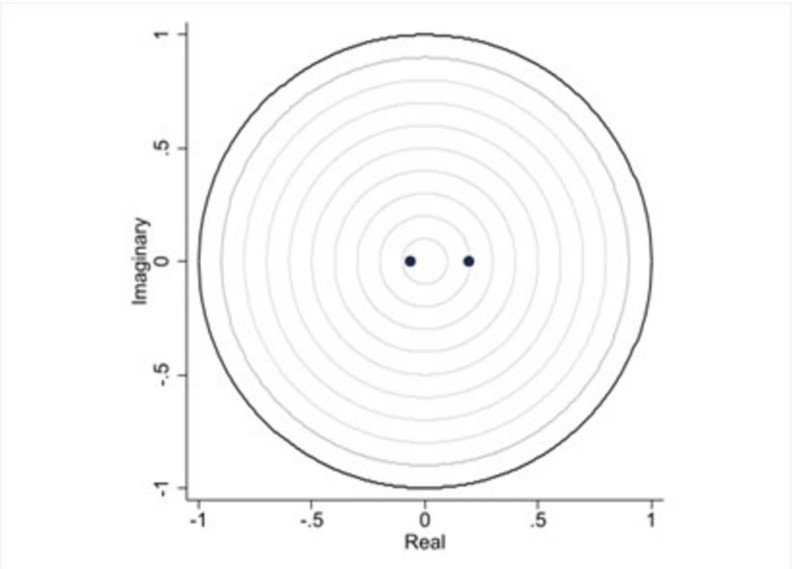

**Fig 2. Roots of the companion matrix.**

on its own has a weak negative effect in the second phase, which returns to positive and converges to zero from the second phase.

The variance decomposition results are consistent with the trend of the pulse graph. The impact of tourism development level and land use efficiency is mainly self-impact(Table 6). In the prediction error variance, the tourism self-effect value is above 0.99, which little affects land use efficiency. In the error of land use efficiency, although the impact on tourism development level is weak, as time goes on, the impact degree increases, the impact degree of the land itself is relatively high, but also with the period later, the degree of influence decreased with a slight change. The first period begins the 12th five-year plan, which strictly controls the land use and protection system, advocates land conservation, and promotes the overall sustainable development of the economy and society. Under policy guidance, major land use projects should reflect the autonomous region's effective development strategies and significant decisions.

**Table 5. Granger causality test.**

| Area | The original hypothesis | Results |
|---|---|---|
| Guangxi | The land is not tour's Grainger reason | Refused |
| | Chi$^2$ | 7.833 |
| | df | 1 |
| | Prob > chi$^2$ | 0.005 |
| | The tour is not land's Grainger reason | Refused |
| | Statistics | 4.475 |
| | df | 1 |
| | Prob > chi$^2$ | 0.034 |

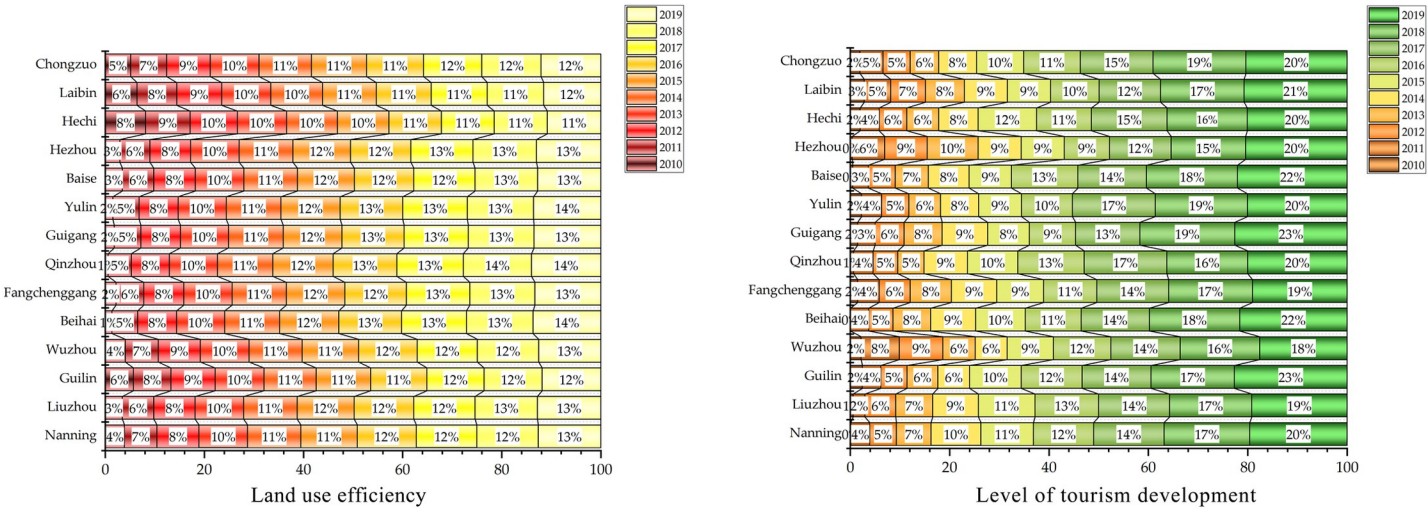

**Fig 3. Land use efficiency and Level of tourism development in Guangxi.**

## 5.4 Discussion

First, we should accurately position the direction of tourism development in Guangxi, and adapt to local conditions, according to different regions to develop tourism and land use

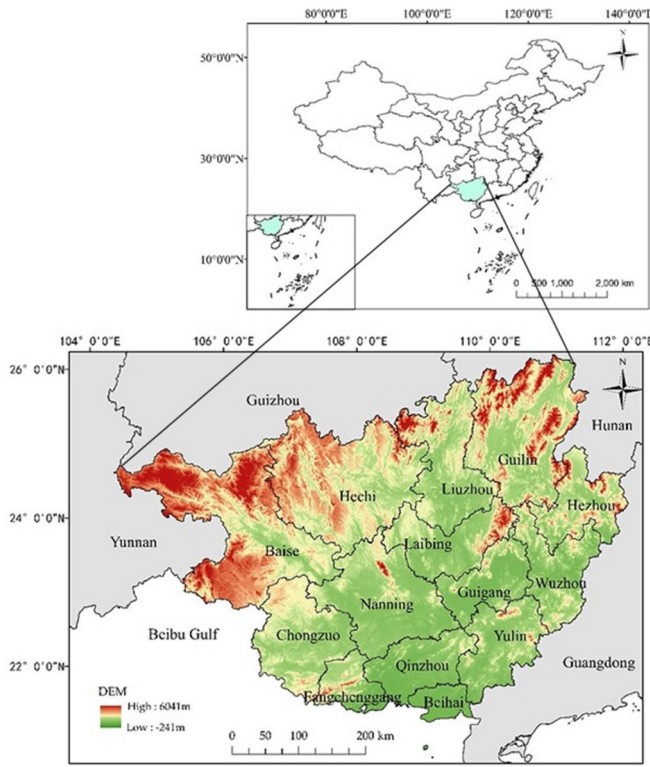

**Fig 4. Topographic map of Guangxi.** Note: Based on the standard map production of the National Natural Resources Department's Standard Map Service website (GS(2019)1822), the base map boundaries have not been modified. (http://bzdt.ch.mnr.gov.cn/download.html?searchText=GS(2019)1822).

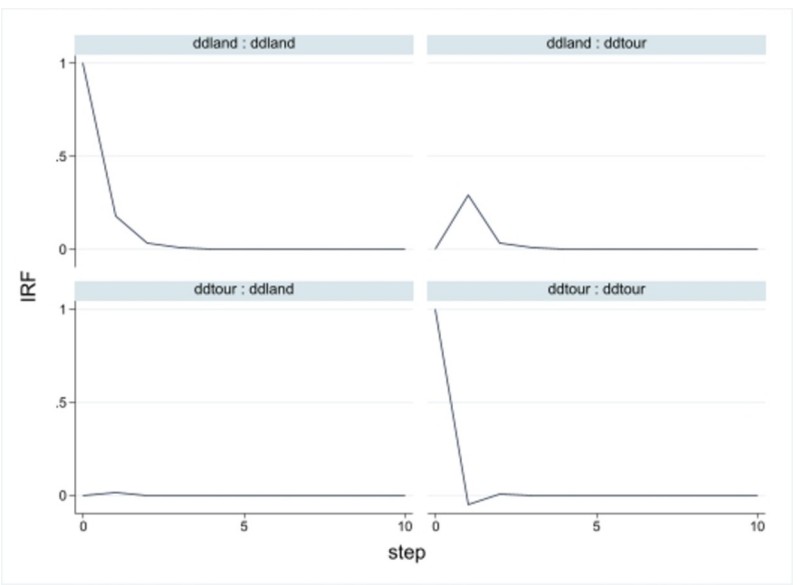

**Fig 5. Impulse response.**

security policies. In previous studies, scholars have found that the grasslands and unused lands in Guangxi have gradually decreased, and there has been a conversion between different types of land use, which has led to a decline in land quality. To improve land use efficiency in mountainous areas, we should focus on spatial distribution, forest protection, cultivated land restoration, and cropland conversion to grassland and forest. Not only that, we should integrate into modern technology, technology management, data management, and so on. Kras, Guangxi province, is mainly a mountainous region. It uses the technology it has mastered to explore further tourism development in high mountain areas, including outdoor tourism, health and wellness tourism, and eco-tourism, to maximize the value of regional land use. The North Sea, Fangchenggang, Qinzhou, and other Gulf of Tonkin coastal areas are close to the coast and rich in tourism resources, so marine tourism development is better. But more attention should be paid to the environmental health of the sea area, water quality, and so on. Discharge sewage, exhaust gas, and solid waste are strictly prohibited. The marine tourism

**Table 6. Results of decomposition of variance.**

| Period (year) | Variance decomposition of tour | | Variance decomposition of land | |
| --- | --- | --- | --- | --- |
| | tour | land | tour | land |
| 0 | 0 | 0 | 0 | 0 |
| 1 | 1 | 0 | 0.093372 | 0.906628 |
| 2 | 0.9995498 | 0.0004501 | 0.1311468 | 0.8688532 |
| 3 | 0.9995423 | 0.0004577 | 0.1318421 | 0.8681579 |
| 4 | 0.9995419 | 0.0004581 | 0.1318755 | 0.8681245 |
| 5 | 0.9995419 | 0.0004581 | 0.1318766 | 0.8681234 |
| 6 | 0.9995419 | 0.0004581 | 0.1318767 | 0.8681234 |
| 7 | 0.9995419 | 0.0004581 | 0.1318767 | 0.8681234 |
| 8 | 0.9995419 | 0.0004581 | 0.1318767 | 0.8681234 |
| 9 | 0.9995419 | 0.0004581 | 0.1318767 | 0.8681234 |
| 10 | 0.9995419 | 0.0004581 | 0.1318767 | 0.8681234 |

environment should be protected, the industrial structure should be optimized, and the discharge rate of the three litters should be reduced. Flat Plains, land use space is more significant and should be based on regional planning major projects such as approval of land use, re-planning, and use of abandoned wasteland. we should promote the high-quality development of tourism and the sustainable use of land in Guangxi. It is conducive to improving people's living environment and raising their living standards. While developing the economy, we should pay attention to protecting and utilizing the ecological environment and land resources.

Second, to promote the high-quality development of tourism, we should improve the service quality and the ecological and cultural environment of scenic spots and make full use of existing resources to create higher and more practical value. The tourism industry belongs to the service industry. The service object is the tourist. However, how to give tourists a greater sense of experience than expected so that tourism destinations obtain a higher evaluation has become a common topic for politicians, entrepreneurs, and scholars. To build tourism projects suitable for developing tourism destinations, improve the service quality of service personnel, and integrate modern technology management so that tourists can more easily enjoy the convenience and sense of experience in tourism. Increase the investment in security measures so that tourists are comfortable and safe to play. Each tourism destination has its orientation, such as parent-child tourism, Health Tourism, holiday tourism, transit tourism, etc. The scenic tourist spots should be developed synthetically by integrating many factors, such as humanity and nature, to subdivide the types of tourists and occupy the leading market share. This has also become the current research and development direction for scholars and tourism companies. All scenic spots should give full play to their existing advantages, optimize the industrial structure, maximize the value of resource utilization, and realize the sustainable development of resource utilization.

## 6. Conclusions

Based on the empirical study of Guangxi tourism development level and land use efficiency, the panel root test, smoothness test, and Granger causality test are carried out with panel data, and the following conclusions are drawn:

1.  Guangxi has a good level of tourism development and a high land use efficiency. The tourism industry is more comprehensive and covers a wide range of fields, blending in the primary industry, secondary sector of the economy, and tertiary sector. Tourism development is beneficial to promote the development of the rural economy, industrial production transformation, and upgrading so that rural tourism and industrial tourism into a new model of change. Tourism development constantly promotes the region's expansion, which encourages the development of the financial industry and adjusts the industrial and labor force structures. However, the saturation of urban land use is more excellent, and suburban and rural areas become the areas to develop new industries. The government adjusts and optimizes rural agricultural land and some industrial land to build public infrastructure.

2.  Tourism development level and land use efficiency interact and promote each other. The development of tourism needs enough impetus to encourage, and adjust the industrial tourism structure, optimize tourism service channels, and improve high-quality economic and social development. Urban land resources are limited, tourism development reaches near saturation, and tourism development needs a particular area of land resources as support. The dynamic development of rural urbanization promotes tourism development in the surrounding suburbs and rural areas. It relies on tourism development to improve living standards, quality of life, and the living environment. On the other hand, tourism development

is limited by the region's area, and the location of available land is less, which adjusts the structure of tourism development and improves the development of high-quality tourism. The limitation of land use has become the driving force of tourism development. At the same time, it can show the level of regional tourism planning, management, and land use rationality.

3. The level of tourism development and land use efficiency is greatly affected by themselves. The pulse graph and variance decomposition results show that tourism is greatly affected by its story and has solid endogenous motivation. Tourism covers a wide range of industries, including the hotel industry, catering industry, stores, scenic spots, and other industries related to each other. It can become a driving force for mutual development. In the same way, the land is affected by its product. Depending on the nature of the land and its geographical location, its intended use and area will have an impact. Capital and labor input is significant, but economic and social value output is small, resulting in low land use value, easily absorbed by other higher input-output planning projects.

This paper explores the correlation and degree of interaction between tourism development level and land use efficiency through research. In future studies, the focus will be on the land planning and utilization of rural tourism, optimizing the rural industrial structure, and researching the demand for tourism development from different types of land. Promoting the development of rural tourism industry with rural land will also be a key area for future research.

## Supporting information

**S1 File.**
(RAR)

## Author Contributions

**Conceptualization:** Qiuli Meng, Hongwen Pi.

**Data curation:** Hongwen Pi.

**Formal analysis:** Qiuli Meng, Lihua Li.

**Investigation:** Qiuli Meng, Hongwen Pi.

**Methodology:** Hongwen Pi.

**Software:** Hongwen Pi.

**Supervision:** Qiuli Meng.

**Writing – original draft:** Qiuli Meng, Hongwen Pi.

**Writing – review & editing:** Qiuli Meng, Hongwen Pi, Tao Xu, Lihua Li.

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
