## [Decision Letter · Decision Letter 0]

31 Aug 2023

PONE-D-23-21805Exploring the Relationship between Regional Tourism Development and Land Use Efficiency: A Case Study of Guangxi Zhuang Autonomous Region, ChinaPLOS ONE

Dear Dr. Pi,

Thank you for submitting your manuscript to PLOS ONE. After careful consideration, we feel that it has merit but does not fully meet PLOS ONE’s publication criteria as it currently stands. Therefore, we invite you to submit a revised version of the manuscript that addresses the points raised during the review process.

ACADEMIC EDITOR: Please attend all the comments given in attached paper along with reviewer comments. ==============================

We look forward to receiving your revised manuscript.

Kind regards,

Nishanthi Rupika Abeynayake, Ph.D

Academic Editor

PLOS ONE

Journal Requirements:

4. We note that [Figures 1 and 4] in your submission contain [map/satellite] images which may be copyrighted. All PLOS content is published under the Creative Commons Attribution License (CC BY 4.0), which means that the manuscript, images, and Supporting Information files will be freely available online, and any third party is permitted to access, download, copy, distribute, and use these materials in any way, even commercially, with proper attribution. For these reasons, we cannot publish previously copyrighted maps or satellite images created using proprietary data, such as Google software (Google Maps, Street View, and Earth). For more information, see our copyright guidelines: http://journals.plos.org/plosone/s/licenses-and-copyright.

a. You may seek permission from the original copyright holder of Figures 1 and 4 to publish the content specifically under the CC BY 4.0 license.  

Additional Editor Comments:

All the comments are in the attached paper

Reviewers' comments:

Reviewer's Responses to Questions

**Comments to the Author**

1. Is the manuscript technically sound, and do the data support the conclusions?

Reviewer #1: Yes

2. Has the statistical analysis been performed appropriately and rigorously? 

Reviewer #1: Yes

3. Have the authors made all data underlying the findings in their manuscript fully available?

Reviewer #1: Yes

4. Is the manuscript presented in an intelligible fashion and written in standard English?

Reviewer #1: Yes

5. Review Comments to the Author

Reviewer #1: Thank you for the opportunity to provide a peer review for this interesting article in “Exploring the Relationship between Regional Tourism Development and Land Use Efficiency: A Case Study of Guangxi Zhuang Autonomous Region, China”

The specific comments for each of the key sections of the article are as follows.

Title:

The title of the article is both well-chosen and appropriate for the content it encompasses. It effectively captures the essence of the research and provides readers with a clear understanding of the article's focus.

Abstract & keywords:

The abstract is poorly structured. Highlights of problem identification need to be included in the abstract. Highlighting the problem identification and stating the objectives of the study clearly will aid to understand the originality of the work.

It is also recommended to include a few keywords.

Introduction:

The introduction sets out a good platform for the study.

However, the identification of the research problem is not well justified. Research questions/objectives need to be clearly set out.

Literature review

This section shows that the authors have a comprehensive understanding of research methods that were used in similar studies and attempted to justify the selection of the SFA model in their study.

However, the literature review lacks theoretical and empirical support on the relationship between land use and tourism development.

Data and Methodology

This section is well-structured and comprehensive.

However, tables and figures are not cited/referred to in the text, and sources of figures and tables are not specified.

Discussion and Conclusions

Well-drawn from the results of the study.

Language and formatting

The quality of English and general presentation is satisfactory.

While the expression is generally clear, there are a few issues in general word formatting, punctuation, and grammar.

A few are marked on the Word document.

6. PLOS authors have the option to publish the peer review history of their article (what does this mean?). If published, this will include your full peer review and any attached files.

Reviewer #1: No

---

## [Author Response · Author response to Decision Letter 0]

22 Sep 2023

First of all, thank you for your valuable comments on this manuscript. We have carefully revised it based on your comments. We have revised them one by one based on your comments and marked them with the "Track Changes" function in Microsoft Word. Because different people may see differ-ent line numbers when using the word format version under "Track Changes" mode. the line number in the response letter refers to the line number in PDF format version. In the following reply report, point-by-point answers to the comments are listed. We hope this revised manuscript has addressed your concerns, and look forward to hearing from you.

 Point 1: Please ensure that your manuscript meets PLOS ONE's style requirements, including those for file naming. The PLOS ONE style templates can be found at https://journals.plos.org/plosone/s/file?id=wjVg/PLOSOne_formatting_sample_main_body.pdf and 

Response 1: Thank you for your reminder. according to the style requirements of PLOS ONE, I have revised the manuscript.

 Point 2: We note that the grant information you provided in the ‘Funding Information’ and ‘Fi-nancial Disclosure’ sections do not match. 

Response 2: Thank you for your reminder. I have verified that there was an error in the 'Finan-cial Disclosure' section and that 'Funding Information' is correct. This article was financially supported by’ the National Office for Philosophy and Social Sciences: Research on Tourism Cultural Space Production and Social Governance Innova-tion in southwest minority traditional villages (20BJY210)’.

 Point 3: We note that you have stated that you will provide repository information for your data at acceptance. Should your manuscript be accepted for publication, we will hold it until you provide the relevant accession numbers or DOIs necessary to access your data. If you wish to make changes to your Data Availability statement, please describe these changes in your cover letter and we will up-date your Data Availability statement to reflect the information you provide.

Response 3: Thank you for your reminder. Yes, I will provide repository information for my data. All data files are available from the Figshare database (doi: https://doi.org/10.6084/m9.figshare.24126672)

 Point 4: We note that [Figures 1 and 4] in your submission contain [map/satellite] images which may be copyrighted. All PLOS content is published under the Creative Commons Attribution License (CC BY 4.0), which means that the manuscript, images, and Supporting Information files will be freely available online, and any third party is permitted to access, download, copy, distribute, and use these materials in any way, even commercially, with proper attribution. For these reasons, we cannot publish previously copyrighted maps or satellite images created using proprietary data, such as Google software (Google Maps, Street View, and Earth). For more information, see our copyright guidelines: http://journals.plos.org/plosone/s/licenses-and-copyright.

Response 4: Thank you for your reminder. I have provided the source for the images in Fig 1 and Fig 4. Which is “Note: Based on the standard map production of the National Natural Resources Department's Standard Map Service website (GS(2019)1822), the base map boundaries have not been modified”. （http://bzdt.ch.mnr.gov.cn/download.html?searchText=GS(2019)1822）

 Point 5: Please review your reference list to ensure that it is complete and correct. If you have cited papers that have been retracted, please include the rationale for doing so in the manuscript text, or remove these references and replace them with relevant current references. Any changes to the refer-ence list should be mentioned in the rebuttal letter that accompanies your revised manuscript. If you need to cite a retracted article, indicate the article’s retracted status in the References list and also in-clude a citation and full reference for the retraction notice.

Response 5: Thank you for your reminder. I have verified that all references in this article are complete and correct and have not been retracted

 Point 6: Revision comments in the manuscript

a:” Its importance to land use.” Is this an incomplete sentence?

Response 6(a): It has been revised. The correct sentence is “This also highlights the im-portance of land use.”

b:” Data Development Envelopment Analysis.” Is this the correct term? Data Development or Data Envelopment Analysis?

Response 6(b): It has been revised. The correct word is “Data Envelopment Analysis”.

---

## [Decision Letter · Decision Letter 1]

4 Dec 2023

PONE-D-23-21805R1Exploring the Relationship between Regional Tourism Development and Land Use Efficiency: A Case Study of Guangxi Zhuang Autonomous Region, ChinaPLOS ONE

Dear Dr. Pi,

Thank you for submitting your manuscript to PLOS ONE. After careful consideration, we feel that it has merit but does not fully meet PLOS ONE’s publication criteria as it currently stands. Therefore, we invite you to submit a revised version of the manuscript that addresses the points raised during the review process.

We look forward to receiving your revised manuscript.

Kind regards,

Jinsheng Jason Zhu

Academic Editor

PLOS ONE

Additional Editor Comments:

I hope this email finds you well. I am writing because your manuscript, "Exploring the Relationship between Regional Tourism Development and Land Use Efficiency: A Case Study of Guangxi Zhuang Autonomous Region, China", which you submitted to Plos One, has been reviewed and a decision has now been made. The reviewers' comments are included at the bottom of this email.

Comments from the Academic Editor:

The paper has excellent data and adopted Panel Data of Guangxi to measure the efficiency of land resources in the context of tourism development. However, this manuscript is still well below a publishable standard, particularly in need of a substantial language edit. Furthermore, the overall structure of the paper need to be further adjusted. Please see the reviewer's comments below. The reviewers would therefore like to see major revisions made to your paper before it can be reconsidered for publication. Please respond to the reviewers' comments and revise your manuscript.

Reviewers' comments:

Reviewer's Responses to Questions

**Comments to the Author**

1. If the authors have adequately addressed your comments raised in a previous round of review and you feel that this manuscript is now acceptable for publication, you may indicate that here to bypass the “Comments to the Author” section, enter your conflict of interest statement in the “Confidential to Editor” section, and submit your "Accept" recommendation.

Reviewer #2: (No Response)

Reviewer #3: (No Response)

2. Is the manuscript technically sound, and do the data support the conclusions?

Reviewer #2: Yes

Reviewer #3: Partly

3. Has the statistical analysis been performed appropriately and rigorously? 

Reviewer #2: Yes

Reviewer #3: Yes

4. Have the authors made all data underlying the findings in their manuscript fully available?

Reviewer #2: Yes

Reviewer #3: No

5. Is the manuscript presented in an intelligible fashion and written in standard English?

Reviewer #2: Yes

Reviewer #3: No

6. Review Comments to the Author

Reviewer #2: The development of the tourism industry typically requires large areas of land for the construction of tourist attractions, hotels, and other related facilities. If the land planning is not reasonable and fails to consider the sustainable use and protection of land, it can result in low land use efficiency. Therefore, studying the relationship between tourism development and land use efficiency holds significant value. The article is well-organized in terms of structure, employs appropriate methods, and the results have practical application value. However, there are some aspects that need improvement:

1. Introduction: It is necessary to supplement and improve the policy background of the research area.

2. Section "3.1 Study Area" should be combined with the topic of the study to describe the overview of land use and tourism development in Guangxi Zhuang Autonomous Region.

3. Section "4.1 Index Construction and Description", the reasons for selecting the indicators should be explained, and if referencing existing literature, it should be properly noted.

4.Section "4.1.2 The index selection of land use efficiency", the rationale for the calculation method of land use efficiency should be explained.

5. Section "4.2.1 Data Processing of regional tourism Indicators." , the meaning of each variable in the formula should be explained.

Reviewer #3: 1. Abstract: This section is written descriptively. It is suggested to include the study findings in a quantitative way.

2. Introduction: (a). This section was written too short and is missing important aspects. (b). The novelty of the study is not properly outlined. (c). Organization of the study is also missing at the end of this section.

3. Literature Review: This section should be divided into two parts (a). The first section must be based on the theoretical foundation followed by the empirical review directly or indirectly related to this study (b). Include up-to-date literature as most of the studies reviewed in this section are old. (c) Furthermore, it is recommended to find out the research gap that this study fills at the end of this section. Please consult papers https://doi.org/10.3390/en14133923;
https://doi.org/10.1007/s10668-020-01023-2;
https://doi.org/10.1007/s10668-022-02194-w for improving and develop this section.

4. Methodology: (a). It is recommended to discuss the significance of each variable included in this study/model (c). Furthermore, the econometric model that examines the impact of independent variables on the dependent variable should be justified (d). It is suggested to discuss how to address different econometric issues in the model.

5. Data analysis and research results: (a). It is recommended to name this section “Results and Discussion” (b).It is recommended to correlate your study with previous studies conducted.

6. It is recommended to include “Future Research Direction” at the end of the manuscript.

7. PLOS authors have the option to publish the peer review history of their article (what does this mean?). If published, this will include your full peer review and any attached files.

Reviewer #2: No

Reviewer #3: **Yes: **Prof. Dr. Dilawar Khan

---

## [Author Response · Author response to Decision Letter 1]

15 Dec 2023

Response to Reviewer #2Comments

First of all, thank you for your valuable comments on this manuscript. We have carefully revised it based on your comments. We have revised them one by one based on your comments and marked them with the "Track Changes" function in Microsoft Word. Because different people may see differ-ent line numbers when using the word format version under "Track Changes" mode. the line number in the response letter refers to the line number in PDF format version. In the following reply report, point-by-point answers to the comments are listed. We hope this revised manuscript has addressed your concerns, and look forward to hearing from you.

Point 1: Introduction: It is necessary to supplement and improve the policy background of the research area.

Response 1: Thank you for your reminder. I added a paragraph to provide the policy back-ground” According to the national policy of 'Three Zones and Three Lines', Guangxi has formu-lated its own 'Three Zones and Three Lines' policy, which aims to improve the national land spatial planning, enhance the quality and value of land utilization, and ensure the reasonable develop-ment of cultural and tourism industries.”

Point 2: Section "3.1 Study Area" should be combined with the topic of the study to describe the overview of land use and tourism development in Guangxi Zhuang Autonomous Region.

Response 2: Thank you for your reminder. I added an overview of the development of tour-ism and land use in Guangxi.” With the increase in cultural tourism land area, the classification management of cultural and tourism land, the adoption of flexible land supply mode, and the re-duction of land costs for cultural and tourism purposes have been achieved. Therefore, it is en-couraged to revitalize and utilize collective rural land, wasteland, etc., increase the area of cultural and tourism land, and improve land reuse efficiency.” 

Delete” Guangxi has become an important region for tourism development due to the influ-ence of the tropical monsoon climate, Karst, and the traditional culture of ethnic minorities.”

Point 3: Section "4.1 Index Construction and Description", the reasons for selecting the indicators should be explained, and if referencing existing literature, it should be properly noted.

Response 3: Thank you for your reminder. I have already annotated the literature used for the selected indicators.

Point 4: Section "4.1.2 The index selection of land use efficiency", the rationale for the calculation method of land use efficiency should be explained.

Response 4: Thank you for your reminder. I have already annotated the literature used for the selected indicators.

I have explained the reasons for the calculation method of land use efficiency.” The develop-ment of society requires the input of labor and capital, with land serving as the carrier of space, and the value generated has certain measurement standards in economic development. Negative out-put that affects social and economic development needs to be eliminated.”

Point 5: Section "4.2.1 Data Processing of regional tourism Indicators.", the meaning of each var-iable in the formula should be explained.

Response 5: Thank you for your reminder. I have added the meanings of the variables in the formula.” Wi, Xi respectively represent the calculation weights of indicators and the level of tourism development in year i.”

Response to Reviewer#3 Comments

First of all, thank you for your valuable comments on this manuscript. We have carefully revised it based on your comments. We have revised them one by one based on your comments and marked them with the "Track Changes" function in Microsoft Word. Because different people may see differ-ent line numbers when using the word format version under "Track Changes" mode. the line number in the response letter refers to the line number in PDF format version. In the following reply report, point-by-point answers to the comments are listed. We hope this revised manuscript has addressed your concerns, and look forward to hearing from you.

Point 1: 1. Abstract: This section is written descriptively. It is suggested to include the study findings in a quantitative way.

Response 1: Thank you for your reminder. I made adjustments to the abstract, and described the conclusions in a quantitative manner.

“There is a reciprocal causation relationship between the Regional tourism development level and land use efficiency in Guangxi, with significant levels of 0.005 and 0.034 respectively, indicat-ing high credibility. This indicates that there is a mutual promotion and interaction between the two, which rely on and drive each other, promoting the joint sustainability of tourism development and land use efficiency”

“The tourism development level is greatly influenced by itself, with impact values all above 0.99. At the same time, land use also has a significant self-impact, with impact values all above 0.87.”

Delete” mutually promote and interact. The two push and rely on each other to cement their relationship. “ , ”Tourism development level and land use efficiency are greatly influenced by themselves”

Point 2: Introduction: (a). This section was written too short and is missing important aspects. (b). The novelty of the study is not properly outlined. (c). Organization of the study is also missing at the end of this section.

Response 2: Thank you for your reminder. The introduction section has been added with the following paragraphs

“Tourism development and construction require a large amount of land for building tourist at-tractions, hotels, and other related facilities. If the land use efficiency in the research area for tour-ism development is not high, it is very likely to damage the land quality and other useful values of the land. Therefore, studying the relationship between the tourism development level and land use efficiency is of great value.”

“According to the national policy of 'Three Zones and Three Lines', Guangxi has formulated its own 'Three Zones and Three Lines' policy, which aims to improve the national land spatial planning, enhance the quality and value of land utilization, and ensure the reasonable develop-ment of cultural and tourism industries.”

“Government policies will continuously address various land use issues that arise in the de-velopment of tourism.”

“Limited land resources have become an important element in the development of tourism.”

“As an important component of tourism resources and a carrier for tourism development, land resources cannot be ignored. Therefore, in the study of the entire tourism development process in Guangxi, our research team further investigated the utilization of tourism land resources.”

Delete” Limited land resources have always been an essential element in the competition for use in ancient and modern times and are now a necessary part of tourism development.”

Point 3: Literature Review: This section should be divided into two parts (a). The first section must be based on the theoretical foundation followed by the empirical review directly or indirectly related to this study (b). Include up-to-date literature as most of the studies reviewed in this section are old. (c) Furthermore, it is recommended to find out the research gap that this study fills at the end of this section. Please consult papers https://doi.org/10.3390/en14133923;
https://doi.org/10.1007/s10668-020-01023-2;
https://doi.org/10.1007/s10668-022-02194-w for improv-ing and develop this section.

Response 3: Thank you for your reminder. The literature review section has been added with the following paragraphs, and references have been made to the literature you recommended.

“Tourism, as a typical form of land use activity, is one of the industries that rely on land re-sources. Land resources are the basic conditions and important resource support for the develop-ment of tourism.”

“Dilawar Khan used SFA to study the relationship between energy efficiency and ecological footprint and believed that efficient utilization of energy resources, as well as investment in agri-culture, are necessary for a sustainable environment.”

“Meanwhile, Professor Dilawar Khan also studied the impact of technological innovation on energy efficiency”

“Zainab Bibi studied the technical and environmental efficiency of the agriculture sector in South Asia and suggested that the agriculture sector should prioritize collaboration in research and development.”

Delete” The high demand for land in early growth, the expansion of urban areas in pursuit of the speed of development, and the relatively high level of waste associated with the increased consumption of land by urban development have once again led to a reflection by scholars and politicians.”

Delete “Meng Tao used the SFA method to study the regional differences in China's tourism industry. The study shows that the regional differences are significant, but the difference value is shrinking.”

Delete “The urban land use efficiency was calculated by the SFA method. Bao Xinzhong be-lieves that the problem of urbanization lies in the contradiction between man and land. The solu-tion to this problem is also the need to realize the sustainable development of cities and [ ] the need to recognize tourism development.”

Point 4: Methodology: (a). It is recommended to discuss the significance of each variable included in this study/model (c). Furthermore, the econometric model that examines the impact of independent variables on the dependent variable should be justified (d). It is suggested to discuss how to address different econometric issues in the model.

Response 4: Thank you for your reminder. The meaning of omitted variables has been added, and other econometric issues have been discussed.

“Wi, Xi respectively represent the calculation weights of indicators and the level of tourism development in year I”

“The PVAR model can analyze the interaction between endogenous variables in panel data, which not only effectively solves the problem of individual heterogeneity, but also fully considers the individual and time effects. This model can solve the econometric problems involved in dif-ferent fields. Therefore”

For this model, variable stationarity has been tested using two methods, Granger causality and stationarity test, in sections 5.2 and 5.1

Point 5: Data analysis and research results: (a). It is recommended to name this section “Results and Discussion” (b).It is recommended to correlate your study with previous studies conducted.

Response 5: Thank you for your reminder. I have changed this section to 'Results and Discus-sion'. and the following content has been added.

” In previous studies, scholars have found that the grasslands and unused lands in Guangxi have gradually decreased, and there has been a conversion between different types of land use, which has led to a decline in land quality”

“We should promote the high-quality development of tourism and the sustainable use of land in Guangxi.”

“This has also become the current research and development direction for scholars and tour-ism companies.”

Point 6: It is recommended to include “Future Research Direction” at the end of the manuscript.

Response 6: Thank you for your reminder. I have added future research directions.” This pa-per explores the correlation and degree of interaction between tourism development level and land use efficiency through research. In future studies, the focus will be on the land planning and utili-zation of rural tourism, optimizing the rural industrial structure, and researching the demand for tourism development from different types of land. Promoting the development of rural tourism industry with rural land will also be a key area for future research.”

---

## [Decision Letter · Decision Letter 2]

2 Jan 2024

Exploring the Relationship between Regional Tourism Development and Land Use Efficiency: A Case Study of Guangxi Zhuang Autonomous Region, China

PONE-D-23-21805R2

Dear Dr. Pi,

We’re pleased to inform you that your manuscript has been judged scientifically suitable for publication and will be formally accepted for publication once it meets all outstanding technical requirements.

Kind regards,

Jinsheng Jason Zhu

Academic Editor

PLOS ONE

**Comments to the Author**

1. If the authors have adequately addressed your comments raised in a previous round of review and you feel that this manuscript is now acceptable for publication, you may indicate that here to bypass the “Comments to the Author” section, enter your conflict of interest statement in the “Confidential to Editor” section, and submit your "Accept" recommendation.

Reviewer #2: All comments have been addressed

Reviewer #3: All comments have been addressed

2. Is the manuscript technically sound, and do the data support the conclusions?

Reviewer #2: Yes

Reviewer #3: Yes

3. Has the statistical analysis been performed appropriately and rigorously? 

Reviewer #2: Yes

Reviewer #3: Yes

4. Have the authors made all data underlying the findings in their manuscript fully available?

Reviewer #2: Yes

Reviewer #3: Yes

5. Is the manuscript presented in an intelligible fashion and written in standard English?

Reviewer #2: Yes

Reviewer #3: Yes

6. Review Comments to the Author

Reviewer #2: (No Response)

Reviewer #3: I appreciate the efforts of the authors for incorporating all my comments and suggestions into the revised version of the manuscript.

7. PLOS authors have the option to publish the peer review history of their article (what does this mean?). If published, this will include your full peer review and any attached files.

Reviewer #2: No

Reviewer #3: No

---

## [Editor Report · Acceptance letter]

10 Jan 2024

PONE-D-23-21805R2 

PLOS ONE

Dear Dr. Pi, 

I'm pleased to inform you that your manuscript has been deemed suitable for publication in PLOS ONE. Congratulations! Your manuscript is now being handed over to our production team.

Kind regards, 

on behalf of

Dr. Jinsheng Jason Zhu 

Academic Editor

PLOS ONE